# Why Learning of Large-Scale Neural Networks Behaves Like Convex Optimization

## Abstract

In this paper, we present some theoretical work to explain why simple gradient descent methods are so successful in solving non-convex optimization problems in learning large-scale neural networks (NN). After introducing a mathematical tool called *canonical space*, we have proved that the objective functions in learning NNs are convex in the canonical model space. We further elucidate that the gradients between the original NN model space and the canonical space are related by a pointwise linear transformation, which is represented by the so-called *disparity* matrix. Furthermore, we have proved that gradient descent methods surely converge to a global minimum of zero loss provided that the disparity matrices maintain full rank. If this full-rank condition holds, the learning of NNs behaves in the same way as normal convex optimization. At last, we have shown that the chance to have singular disparity matrices is extremely slim in large NNs. In particular, when over-parameterized NNs are randomly initialized, the gradient decent algorithms converge to a global minimum of zero loss in probability.

## 1 Introduction

In the past decade, deep learning methods have been successfully applied to many challenging real-world applications, ranging from speech recognition to image classification and to machine translation, and more. These successes largely rely on learning a very large neural network from a plenty of labelled training samples. It is well known that the objective functions of nonlinear neural networks are non-convex, and even non-smooth when the popular ReLU activation functions are used. The traditional optimization theory has regarded this type of high-dimensional non-convex optimization problem as infeasible to solve (Blum & Rivest, 1992; Auer et al., 1995) and any gradient-based first-order local search methods do not converge to any global optimum in probability (Nesterov, 2004). However, the deep learning practices in the past few years have seriously challenged what is suggested by the optimization theory. No matter what structures are used in a large scale neural network, either feed-forward or recurrent, either convolutional or fully-connected, either ReLU or sigmoid, the simple first-order methods such as stochastic gradient descent and its variants can consistently converge to a global minimum of zero loss no matter what type of labelled training samples are used. At present, one of the biggest mysteries in deep learning is how to explain why the learning of NNs is unexpectedly easy to solve. Plenty of empirical evidence accumulated over the past years in various domains has strongly suggested that there must be some fundamental reasons in theory to guarantee such consistent convergence to global minimum in learning of large-scale neural networks. Recently, lots of empirical work (Goodfellow et al., 2014; Zhang et al., 2015) and theoretical analysis (Baldi & Hornik, 1989; Choromanska et al., 2014; Livni et al., 2014; Kawaguchi, 2016; Safran & Shamir, 2016; Soudry & Carmon, 2016; Nguyen & Hein, 2017; Chizat & Bach, 2018; Safran & Shamir, 2018; Du et al., 2018b;a; Allen-Zhu et al., 2018; Zou et al., 2018) have been reported to tackle this question from many different aspects.

In this paper, we present a novel theoretical analysis to uncover the mystery behind the learning of neural networks. Comparing with all previous theoretical work, such as Choromanska et al. (2014); Kawaguchi (2016); Soudry & Carmon (2016); Du et al. (2018b); Zou et al. (2018), we use some unique mathematical tools, like canonical model space and Fourier analysis, to derive theoretical proofs under a very general setting without any unrealistic assumption on the model structure and data distribution. Unlike many math-heavy treatments in the literature, our method is technically concise and conceptually intuitive so that it leads to an intelligible sufficient condition for such

consistent convergence to occur. Our theoretical results have also well explained many common practices widely followed by deep learning practitioners.

## 2 PROBLEM FORMULATION

In this work, we study model estimation problems under the standard machine learning setting. Given a finite training set of $T$ samples of input and output pairs, denoted as $\mathcal{D}_T = \{(\mathbf{x}_1, y_1), (\mathbf{x}_2, y_2), \cdots, (\mathbf{x}_T, y_T)\}$, the goal is to learn a model from input to output over the entire feature space $f : \mathbf{x} \to y$ ($\mathbf{x} \in \mathbb{R}^K, y \in \mathbb{R}$), which will be used to predict future inputs. In all practical applications, the input $\mathbf{x}$ normally lies in a constrained region in $\mathbb{R}^K$, which can always be normalized into a unit hypercube, denoted as $\mathbb{U}_K \triangleq [0, 1]^K$. Without losing generality, we may formulate the above learning problem as to search for the optimal function $f(\mathbf{x})$ within the function class $L^1(\mathbb{U}_K)$ to minimize a loss functional measured in $\mathcal{D}_T$, where $L^1(\mathbb{U}_K)$ denotes all bounded absolutely integrable functions defined in $[0, 1]^K$. The loss function is computed as empirical errors accumulated over all training samples in $\mathcal{D}_T$. The empirical error at each sample, $(\mathbf{x}_t, y_t)$, is usually measured by a convex loss function $l(y_t, f(\mathbf{x}_t))$, which penalizes mis-classification errors and rewards correct classifications. In other words, a meaningful loss function $l(y, y')$ always satisfies:

$$l(y, y') \Rightarrow \begin{cases} = 0 & (y = y') \\ > 0 & (y \neq y') \end{cases} . \tag{1}$$

Obviously, the popular loss measures, such as mean squared error, cross-entropy, always satisfy the condition in eq.(1) and are clearly convex functions with respect to its second argument. Therefore, we may use a general notation to represent this learning problem as follows:

$$f^* = \arg \min_{f \in L^1(\mathbb{U}_K)} Q(f|\mathcal{D}_T) = \arg \min_{f \in L^1(\mathbb{U}_K)} \sum_{t=1}^{T} l(y_t, f(\mathbf{x}_t)) \tag{2}$$

Next, let's consider how to parametarize the function space $L^1(\mathbb{U}_K)$ to make the above optimization computationally feasible.

### 2.1 LITERAL MODEL SPACE

Based on the universal approximation theorems in Cybenko (1989); Hornik et al. (1989); Hornik (1991), any function in $L^1(\mathbb{U}_K)$ can be approximated up to arbitrary precision by a well-structured neural network of sufficiently large model size. Here, we use $\Lambda_M$ to represent the set of all well-structured neural networks using $M$ free model parameters, and all weight parameters of each neural network are denoted as a vector $\mathbf{w}$ ($\mathbf{w} \in \mathbb{R}^M$). Obviously, if $M$ is made sufficiently large, $\Lambda_M$ is a good choice to parameterize the function space $L^1(\mathbb{U}_K)$ because given any bounded function $f(\mathbf{x}) \in L^1(\mathbb{U}_K)$, there exists at least one set of model parameters in $\Lambda_M$, denoted as $\mathbf{w}$, to make the corresponding neural network approximate $f(\mathbf{x})$ up to arbitrary precision. In this case, the function represented by the underlying neural network is denoted as $f_\mathbf{w}(\mathbf{x})$. Furthermore, if $M$ is finite and all model parameters are bounded, every function represented by each possible neural network in $\Lambda_M$ belongs to $L^1(\mathbb{U}_K)$. As a result, once we pre-determine the neural network structure of model size $M = |\mathbf{w}|$ (assume that $M$ is sufficiently large), the functional minimization problem in eq.(2) can be simplified into an equivalent parameter optimization problem as follows:

$$\mathbf{w}^* = \arg \min_{\mathbf{w} \in \mathbb{R}^M} Q(f_\mathbf{w}|\mathcal{D}_N) = \arg \min_{\mathbf{w} \in \mathbb{R}^M} \sum_{t=1}^{T} l(y_t, f_\mathbf{w}(\mathbf{x}_t)) . \tag{3}$$

However, due to the weight-space symmetries and network redundancy, the mapping from $\Lambda_M$ to $L^1(\mathbb{U}_K)$ is surjective but not injective. For any function $f \in L^1(\mathbb{U}_K)$, there always exist many different $\mathbf{w}$ in $\Lambda_M$ to make neural networks represent $f$ equally well. In this work, the space of neural networks, $\Lambda_M$, is called *literal model space* of $L^1(\mathbb{U}_K)$ because it only satisfies the existence requirement but not the uniqueness one. It is the non-uniqueness of $\Lambda_M$ that makes it extremely difficult to directly conduct the theoretical analysis for the optimization problem in eq.(3).

## 2.2 CANONICAL MODEL SPACE

Here, let's consider the so-called *canonical model space* for $L^1(\mathbb{U}_K)$. A model space is called to be *canonical* if it satisfies both existence and uniqueness requirements. In other words, for every function $f \in L^1(\mathbb{U}_K)$, there exists exactly one unique model in the *canonical model space* to represent $f$. Moreover, every model in the canonical model space corresponds to a unique function in $L^1(\mathbb{U}_K)$. Therefore, the mapping between $L^1(\mathbb{U}_K)$ and its canonical model space is bijective. For $L^1(\mathbb{U}_K)$, the multivariate Fourier series naturally form such a canonical model space for $L^1(\mathbb{U}_K)$.

As in Pinsky (2002), given any function $f(\mathbf{x}) \in L^1(\mathbb{U}_K)$, we may compute its multivariate Fourier coefficients as follows:

$$\theta_{\boldsymbol{k}} = \int \cdots \int_{\mathbf{x} \in \mathbb{U}_K} f(\mathbf{x})\, e^{-2\pi i \boldsymbol{k} \cdot \mathbf{x}} d\mathbf{x} \quad (\forall \boldsymbol{k} \in \mathbb{Z}^K) \tag{4}$$

where $i = \sqrt{-1}$ and $\theta_{\boldsymbol{k}} \in \mathbb{C}$, and $\boldsymbol{k} = [k_1, k_2, \cdots, k_K]$ is a tuple of $K$ integers, denoting the $K$-dimensional index of each Fourier coefficient. All Fourier coefficients of a function $f(\mathbf{x})$ can be arranged as an infinite sequence, i.e. $\boldsymbol{\theta} = \{\theta_{\boldsymbol{k}} \mid \boldsymbol{k} \in \mathbb{Z}^K\}$, which in turn can be viewed as a point in a Hilbert space with an infinite number of dimensions. This Hilbert space is denoted as $\boldsymbol{\Theta}$. For the notational simplicity, we may represent eq.(4) by a generic mapping from any $f(\mathbf{x}) \in L^1(\mathbb{U}_K)$ to a $\boldsymbol{\theta} \in \boldsymbol{\Theta}$ as: $\boldsymbol{\theta} := \mathscr{F}\big(f(\mathbf{x})\big)$.

According to the Fourier theorem, for any function $f(\mathbf{x}) \in L^1(\mathbb{U}_K)$, these Fourier coefficients may be used to perfectly reconstruct $f(\mathbf{x})$ by summing the following Fourier series, which converges everywhere in $\mathbf{x} \in \mathbb{U}_K$:

$$f(\mathbf{x}) = \sum_{\boldsymbol{k} \in \mathbb{Z}^K} \theta_{\boldsymbol{k}}\, e^{2\pi i \boldsymbol{k} \cdot \mathbf{x}} := \mathscr{F}^{-1}(\mathbf{x}|\boldsymbol{\theta}). \tag{5}$$

Because both eq.(4) and eq.(5) converge for any function $f(\mathbf{x}) \in L^1(\mathbb{U}_K)$, therefore, $\boldsymbol{\Theta}$ is a canonical model space of $L^1(\mathbf{x})$. Every function in $f(\mathbf{x})$ can be uniquely represented by all of its Fourier coefficients $\boldsymbol{\theta}$ in $\boldsymbol{\Theta}$.

According to the Riemann-Lebesgue lemma (pp.18 in Pinsky (2002)), the Fourier coefficients $\theta_{\boldsymbol{k}}$ decay from both sides in every dimension of $\boldsymbol{k}$, i.e., $|\theta_{\boldsymbol{k}}| \to 0$ as the absolute value of any dimension of $\boldsymbol{k}$ goes to infinity, $\max |\boldsymbol{k}| \to \infty$. As a result, the Fourier serie in eq.(5) can be truncated into a partial sum centered at the origin. Given any small number $\epsilon > 0$, for any function $f(\mathbf{x}) \in L^1(\mathbb{U}_K)$, we can always choose a finite number of the most significant Fourier coefficients located in the center, which are denoted as $\mathcal{N}_\epsilon := \{\boldsymbol{k} \mid |k_1| \leq N_1, \cdots, |k_K| \leq N_K\}$. The cardinality of $\mathcal{N}_\epsilon$ is denoted as $N = |\mathcal{N}_\epsilon|$. The truncated Fourier coefficients can also be arranged onto an $N$-dimensional vector, $\boldsymbol{\theta}_\epsilon = \{\theta_{\boldsymbol{k}} \mid \boldsymbol{k} \in \mathcal{N}_\epsilon\}$, which may be viewed as a point in a complete normed linear space with $N$ dimensions. This finite-dimensional normed linear space is thus denoted as $\boldsymbol{\Theta}_\epsilon$. Moreover, we may use these truncated coefficients in $\mathcal{N}_\epsilon$ to form a partial sum to approximate the original function $f(\mathbf{x})$ up to the precision $1 - \epsilon^2$:

$$\hat{f}(\mathbf{x}) = \sum_{\boldsymbol{k} \in \mathcal{N}_\epsilon} \theta_{\boldsymbol{k}}\, e^{i \boldsymbol{k} \cdot \mathbf{x}} \tag{6}$$

where $\hat{f}(\mathbf{x})$ is an infinitely differentiable function in $\mathbb{U}_K$, approximating $f(\mathbf{x})$ up to $1 - \epsilon^2$ in the $L^2$ norm, i.e. $\int \cdots \int_{\mathbf{x} \in \mathbb{U}_K} \|f(\mathbf{x}) - \hat{f}(\mathbf{x})\|^2 d\mathbf{x} \leq \epsilon^2$.

In summary, for any function $f(\mathbf{x}) \in L^1(\mathbb{U}_K)$, we may calculate its Fourier coefficients as in eq.(4) to represent it uniquely in the infinite-dimensional canonical space $\boldsymbol{\Theta}$ as $\boldsymbol{\theta} = \mathscr{F}(f(\mathbf{x}))$ where $\boldsymbol{\theta} \in \boldsymbol{\Theta}$. Furthermore, for the computational convenience, we may approximate $\boldsymbol{\Theta}$ with a finite-dimension space up to any arbitrary precision. For a given tolerance error $\epsilon$, we may truncate $\boldsymbol{\theta}$ by leaving out all insignificant coefficients and construct $\boldsymbol{\theta}_\epsilon$ of $N$ dimensions as above. This process is conveniently denoted as a mapping: $\boldsymbol{\theta}_\epsilon = \mathscr{F}_\epsilon(f(\mathbf{x}))$. On the other hand, $\boldsymbol{\theta}_\epsilon$ may be used to construct $\hat{f}(\mathbf{x})$ based on the partial sum in eq. (6), i.e. $\hat{f}(\mathbf{x}) = \mathscr{F}^{-1}(\mathbf{x}|\boldsymbol{\theta}_\epsilon)$. As shown above, $\hat{f}(\mathbf{x})$ approximates the original function $f(\mathbf{x})$ up to the precision $1 - \epsilon^2$, denoted as $\hat{f}(\mathbf{x}) = f(\mathbf{x}) + O(\epsilon)$. Here, we call this finite dimensional space $\boldsymbol{\Theta}_\epsilon$ as an $\epsilon$-precision canonical model space of $L^1(\mathbb{U}_K)$. Obviously, the approximation error $\epsilon^2$ can be made arbitrarily small so as to make $\boldsymbol{\Theta}_\epsilon$ approach $\boldsymbol{\Theta}$ as much as possible.

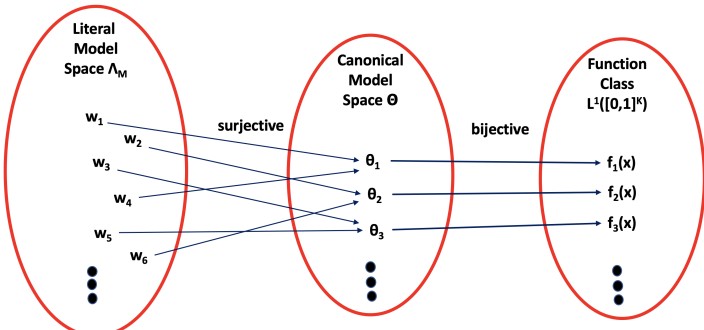

Figure 1: An illustration to show how literal vs. canonical model spaces are related when neural networks are used to approximate function class $L^1(\mathbb{U}_K)$.

## 2.3 LITERAL VS. CANONICAL MODEL SPACE

For any neural network in the literal model space $\Lambda_M$, assuming its model parameter is $\mathbf{w}^{(0)}$, it forms a function $f_{\mathbf{w}^{(0)}}(\mathbf{x})$ between its input and output. As we know $f_{\mathbf{w}^{(0)}}(\mathbf{x}) \in L^1(\mathbb{U}_K)$, we can easily find its corresponding representation in the canonical model space with infinite dimensions via $\boldsymbol{\theta}^{(0)} = \mathscr{F}(f_{\mathbf{w}^{(0)}}(\mathbf{x}))$, where $\boldsymbol{\theta}^{(0)} \in \boldsymbol{\Theta}$. Similarly, we may truncate $\boldsymbol{\theta}^{(0)}$ to find its representation in an $\epsilon$-precision canonical model space with $N$ dimensions as $\boldsymbol{\theta}_\epsilon^{(0)} = \mathscr{F}_\epsilon(f_{\mathbf{w}^{(0)}}(\mathbf{x}))$, where $\boldsymbol{\theta}_\epsilon^{(0)} \in \boldsymbol{\Theta}_\epsilon$. As shown in Figure 1, the mapping from $\Lambda_M$ to $\boldsymbol{\Theta}$ is surjective but not injective while the mapping from $\boldsymbol{\Theta}$ to $L^1(\mathbb{U}_K)$ is bijective.

## 3 MAIN RESULTS

### 3.1 LEARNING IN CANONICAL SPACE

Let's consider the original learning problem eq.(2) in the canonical model space. Since canonical model spaces normally have nice structures, i.e., every $f(\mathbf{x})$ is uniquely represented as a linear combination of some fixed orthogonal base functions. The functional minimization in eq.(2) turns to be an optimization problem to determine the unknown coefficients in a linear combination as follows:

$$\boldsymbol{\theta}^* = \arg \min_{\boldsymbol{\theta} \in \mathbb{C}^{|\boldsymbol{\Theta}|}} Q(\boldsymbol{\theta}|\mathcal{D}_T) = \arg \min_{\boldsymbol{\theta} \in \mathbb{C}^{|\boldsymbol{\Theta}|}} \sum_{t=1}^{T} l\left(y_i, \mathscr{F}^{-1}(\mathbf{x}_i \,|\, \boldsymbol{\theta})\right) \tag{7}$$

This learning problem turns out to be strikingly easy in the canonical model space.

**Theorem 1** *The objective function $Q(f|\mathcal{D}_T)$ in eq.(2), when constructed using a convex loss measure $l(\cdot)$, is a convex function in the canonical model space. If the dimensionality of the canonical space is not less than the number of training samples in $\mathcal{D}_T$, the global minimum achieves zero loss.*

**Proof:** Given an arbitrarily small tolerance error $\epsilon \geq 0$, any function $f(\mathbf{x}) \in L^1(\mathbb{U}_K)$ can be uniquely mapped to a point in an canonical space $\boldsymbol{\Theta}_\epsilon$ as: $\boldsymbol{\theta}_\epsilon = \mathscr{F}_\epsilon(f(\mathbf{x}))$. Meanwhile, the function $f(\mathbf{x})$ may be approximated by $\boldsymbol{\theta}_\epsilon = \{\boldsymbol{k} \mid \boldsymbol{k} \in \mathcal{N}_\epsilon\}$ with the partial sum of Fourier series: $f_{\boldsymbol{\theta}_\epsilon}(\mathbf{x}) = \sum_{\boldsymbol{k} \in \mathcal{N}_\epsilon} \theta_{\boldsymbol{k}} \, e^{2\pi i \boldsymbol{k} \cdot \mathbf{x}} = \sum_{\boldsymbol{k} \in \mathcal{N}_\epsilon} \theta_{\boldsymbol{k}} \eta_{\boldsymbol{k}}(\mathbf{x})$, where $\eta_{\boldsymbol{k}}(\mathbf{x}) = e^{2\pi i \boldsymbol{k} \cdot \mathbf{x}}$. If we constrain to optimize eq.(2) in the canonical space $\boldsymbol{\Theta}_\epsilon$, the objective function is represented as:

$$Q(f_{\boldsymbol{\theta}_\epsilon}|\mathcal{D}_T) = \sum_{t=1}^{T} l\left(y_t, f_{\boldsymbol{\theta}_\epsilon}(\mathbf{x}_t)\right) = \sum_{t=1}^{T} l\left(y_t, \sum_{\boldsymbol{k} \in \mathcal{N}_\epsilon} \theta_{\boldsymbol{k}} \eta_{\boldsymbol{k}}(\mathbf{x}_t)\right).$$

Since the loss measure $l(\cdot)$ itself is a convex function of its second argument, and the argument is a linear combination of all parameters $\theta_{\boldsymbol{k}}$, therefore, the right-hand side of the above equation is a

convex function of its parameter $\boldsymbol{\theta}_\epsilon$. When mean squared error is used for $l(\cdot)$, the above procedure is the same as the well-known least square method.

Due to eq.(1), if a global minimum $\boldsymbol{\theta}_\epsilon^*$ achieves zero loss, it must satisfy: $l(y_i, f(\mathbf{x}_i)) = l\left(y_i, \sum_{\boldsymbol{k} \in \mathcal{N}_\epsilon} \theta_{\boldsymbol{k}} \eta_{\boldsymbol{k}}(\mathbf{x}_i)\right) = 0 \ (i = 1, 2, \cdots, T)$, which are equivalent to a system of $T$ linear equations: $\sum_{\boldsymbol{k} \in \mathcal{N}_\epsilon} \theta_{\boldsymbol{k}} \eta_{\boldsymbol{k}}(\mathbf{x}_i) = y_i \ (i = 1, 2, \cdots, T)$.

Obviously, if the dimensionality of the canonical model space is not less than the number of point-wise distinct (and noncontradictory) training samples, $T$, we have more free variables in $\boldsymbol{\theta}_\epsilon^*$ than the total number of linear equations. Therefore, there exists at least one solution to jointly satisfy these independent equations, which also achieves the zero loss at the same time. If the dimensionality of the canonical model space is equal to the number of point-wise distinct training samples, there exists a unique global minimum that achieves zero loss. ∎

**Corollary 1** $Q(f|\mathcal{D}_T)$ *in eq.(2) is a convex functional in* $L^1(\mathbb{U}_K)$.

**Proof:** For any two functions, $f_1, f_2 \in L^1(\mathbb{U}_K)$, we may map them into the canonical model space $\boldsymbol{\Theta}$ to find their representations as: $\boldsymbol{\theta}_1 = \mathscr{F}(f_1)$ and $\boldsymbol{\theta}_2 = \mathscr{F}(f_2)$. As shown in eq.(5), both $f_1$ and $f_2$ can be represented as a linear function of $\boldsymbol{\theta}_1$ and $\boldsymbol{\theta}_2$. Since $l(\cdot)$ is a convex function, for any $0 \leq \varepsilon \leq 1$, it is trivial to show that $Q(\varepsilon f_1 + (1-\varepsilon) f_2|\mathcal{D}_T) \leq \varepsilon Q(f_1|\mathcal{D}_T) + (1-\varepsilon) Q(f_2|\mathcal{D}_T)$. Therefore, $Q(f|\mathcal{D}_T)$ is a convex functional in $L^1(\mathbb{U}_K)$. ∎

### 3.2 GOING BACK TO LITERAL SPACE FROM CANONICAL SPACE

As shown above, learning in the canonical model space is a standard convex optimization problem. However, the direct learning in the canonical model space may be computationally prohibitive in high-dimensional canonical spaces. At present, the common practice in machine learning is still to learn neural networks in the literal model space. Here we will look at how the canonical space may help to understand the learning behaviours of neural networks in the literal space.

The learning of neural networks mainly uses the first-order methods, which solely rely on the gradients of the objective function. We will first investigate how the gradients in the literal model space are related to the gradients in the canonical model space. Given model parameters $\mathbf{w}$ of a neural network, which may be viewed as a point in the literal space $\Lambda_M$ with $M = |\mathbf{w}|$ dimensions, the objective function at $\mathbf{w}$ is denoted as $Q(f_\mathbf{w}|\mathcal{D}_T)$. If the training set $\mathcal{D}_T$ has $T$ point-wise distint training samples, we consider an $\epsilon$-precision canonical space, $\boldsymbol{\Theta}_\epsilon$. The cardinality of $\boldsymbol{\Theta}_\epsilon$, denoted as $N$, is chosen to satisfy two conditions: i) $N$ is not smaller than $T$; ii) $N$ is large enough to make the truncation error $\epsilon$ sufficiently small. In this way, for any $\mathbf{w}$ in $\Lambda_M$, it may be mapped to this canonical space as $\boldsymbol{\theta}_\epsilon = \mathscr{F}_\epsilon(f_\mathbf{w}(\mathbf{x})) \ (\boldsymbol{\theta}_\epsilon \in \boldsymbol{\Theta}_\epsilon)$. Obviously, $Q(f_\mathbf{w}|\mathcal{D}_N) = Q(\boldsymbol{\theta}|\mathcal{D}_T) + O(\epsilon)$. However, the residual error term may be ignored since it can be made arbitrarily small when we choose a large enough $N$. Moreover, due to $N \geq T$, the global minimum of the objective function achieves zero loss in this canonical space. In this work, we consider the learning problem of over-parameterized neural networks, where the dimensionality of the literal space, $M = |\mathbf{w}|$, is much larger than the dimensionality of this canonical space, namely $M \gg N$.

Based on the chain rule, we have:

$$\nabla_\mathbf{w} Q(f_\mathbf{w}|\mathcal{D}_T) = \nabla_{\boldsymbol{\theta}_\epsilon} Q(f_{\boldsymbol{\theta}_\epsilon}|\mathcal{D}_T) \nabla_\mathbf{w} \boldsymbol{\theta}_\epsilon = \nabla_{\boldsymbol{\theta}_\epsilon} Q(f_{\boldsymbol{\theta}_\epsilon}|\mathcal{D}_T) \nabla_\mathbf{w} \mathscr{F}_\epsilon(f_\mathbf{w}(\mathbf{x}))$$

which can be represented as the following matrix format:

$$\begin{bmatrix} \frac{\partial Q}{\partial w_1} \\ \vdots \\ \frac{\partial Q}{\partial w_m} \\ \vdots \\ \frac{\partial Q}{\partial w_M} \end{bmatrix}_{M \times 1} = \begin{bmatrix} \mathscr{F}_\epsilon\left(\frac{\partial f_\mathbf{w}(\mathbf{x})}{\partial w_1}\right) \\ \vdots \\ \mathscr{F}_\epsilon\left(\frac{\partial f_\mathbf{w}(\mathbf{x})}{\partial w_m}\right) \\ \vdots \\ \mathscr{F}_\epsilon\left(\frac{\partial f_\mathbf{w}(\mathbf{x})}{\partial w_M}\right) \end{bmatrix}_{M \times N} \begin{bmatrix} \frac{\partial Q}{\partial \boldsymbol{\theta}_1} \\ \vdots \\ \frac{\partial Q}{\partial \boldsymbol{\theta}_n} \\ \vdots \\ \frac{\partial Q}{\partial \boldsymbol{\theta}_N} \end{bmatrix}_{N \times 1} . \tag{8}$$

where the $M \times N$ matrix is called *disparity* matrix, denoted as $\mathbf{H}(\mathbf{w})$. The *disparity* matrix $\mathbf{H}(\mathbf{w})$ is composed of $M$ sets of Fourier coefficients (computed over the input $\mathbf{x}$) of partial derivatives of

the neural network function with respect to each of its weights. When we form the $m$-th row of the *disparity* matrix, we first compute the partial derivative of the neural network function $f_{\mathbf{w}}(\mathbf{x})$ with respect to $m$-th parameter, $w_m$, in the neural netowork. This partial derivative, $\frac{\partial f_{\mathbf{w}}(\mathbf{x})}{\partial w_m}$, is still a function of input $\mathbf{x}$. Then we first apply Fourier series in eq.(5) to it and then truncate to keep the total $N$ most significant Fourier coefficients. These $N$ coefficients are aligned to be placed in the $m$-th row of the disparity matrix $\mathbf{H}(\mathbf{w})$. This process is repeated for every parameter of the neural network so as to fill up all rows in the $M \times N$ matrix. Obviously, the *disparity* matrix depends on $\mathbf{w}$ because when $\mathbf{w}$ take different values, these function derivatives are normally different and so are their Fourier coefficients. For any given neural network $\mathbf{w}$, this mapping can be represented as a compact matrix format:

$$\Big[\nabla_{\mathbf{w}}Q\Big]_{M \times 1} = \Big[\mathbf{H}(\mathbf{w})\Big]_{M \times N}\Big[\nabla_{\boldsymbol{\theta}_\epsilon}Q\Big]_{N \times 1}. \tag{9}$$

From the above, we can see that the gradient in the literal space is related to its corresponding gradient in the canonical space through a linear transformation at every $\mathbf{w}$, which is represented by the *disparity* matrix $\mathbf{H}(\mathbf{w})$. The *disparity* matrix varies when $\mathbf{w}$ moves from one model to another in the literal space. As a result, we say that the gradient in the literal space is linked to its corresponding gradient in the canonical space via a point-wise linear transformation.

**Lemma 1** *Assume the used neural network is sufficiently large ($M \geq N$). If $\mathbf{w}^*$ is a stationary point of the objective function $Q(f_{\mathbf{w}}|\mathcal{D}_T)$ and the corresponding disparity matrix $\mathbf{H}(\mathbf{w}^*)$ has full rank at $\mathbf{w}^*$, then $\mathbf{w}^*$ is a global minimum of $Q(f_{\mathbf{w}}|\mathcal{D}_T)$.*

**Proof:** If $\mathbf{w}^*$ is a stationary point, it implies that the gradient vanishes at $\mathbf{w}^*$ in the literal model space, i.e., $\Big[\nabla_{\mathbf{w}}Q(\mathbf{w}^*)\Big]_{M \times 1} = 0$. Substituting this into eq.(9), we have

$$\Big[\mathbf{H}(\mathbf{w}^*)\Big]_{M \times N}\Big[\nabla_{\boldsymbol{\theta}_\epsilon}Q(\boldsymbol{\theta}_\epsilon)\Big]_{N \times 1} = 0 \tag{10}$$

because $M \geq N$ and $\mathbf{H}(\mathbf{w}^*)$ has the full rank ($N$), the only solution to the above equations in the canonical space is $\nabla_{\boldsymbol{\theta}_\epsilon}Q(\boldsymbol{\theta}_\epsilon^*) = 0$. In other words, the corresponding model $\boldsymbol{\theta}_\epsilon^*$ in the canonical space is also a stationary point. Since $Q(\boldsymbol{\theta}_\epsilon)$ is a convex function in the canonical space, this implies the corresponding model $\boldsymbol{\theta}^*$ is a global minimum. Due to the fact that the objective functions are equal across two spaces when we map $\mathbf{w}^*$ to $\boldsymbol{\theta}_\epsilon^*$, i.e., $Q(\mathbf{w}^*) = Q(\boldsymbol{\theta}_\epsilon^*)$, we conclude that $\mathbf{w}^*$ is also a global minimum in the literal space. ∎

**Lemma 2** *If $\mathbf{w}^{(0)}$ is a stationary point of $Q(f_{\mathbf{w}}|\mathcal{D}_T)$, but the corresponding disparity matrix $\mathbf{H}(\mathbf{w}^{(0)})$ does not have full rank at $\mathbf{w}^{(0)}$, then $\mathbf{w}^{(0)}$ may be a local minimum or saddle point or global minimum of $Q(f_{\mathbf{w}}|\mathcal{D}_T)$.*

**Proof:** The *disparity* matrix is an $M \times N$ matrix and we still assume $M \geq N$. If $\mathbf{H}(\mathbf{w}^{(0)})$ degenerates and does not have full rank at $\mathbf{w}^{(0)}$, it is possible to have zero or some nonzero solutions in the canonical space to the system of under-specified equations in eq.(10). For the zero solution $\nabla_{\boldsymbol{\theta}_\epsilon}Q(\boldsymbol{\theta}_\epsilon^{(0)}) = 0$, the corresponding model of $\boldsymbol{\theta}_\epsilon^{(0)}$ in literal space is a global minimum because the objective function is convex in the canonical space. However, for the non-zero solutions, $\nabla_{\boldsymbol{\theta}_\epsilon}Q(\bar{\boldsymbol{\theta}}_\epsilon^{(0)}) \neq 0$, the corresponding model of $\bar{\boldsymbol{\theta}}_\epsilon^{(0)}$ in literal space is certainly not a global minimum since the gradient does not vanish in the canonical space. They may corresponds to some bad local minimum or saddle points in the literal space since the gradient vanishes **only** in the literal space. ∎

As we know that neural networks are directly learned in the literal space as in eq.(3). We normally use some fairly simple first-order local search methods to solve this non-convex optimization problem. These methods include gradient descent (GD) and more computationally efficient stochastic gradient descent (SGD) algorithms. As shown in Algorithm 1, these algorithms operate in a pretty simple fashion: it starts from a random model and the model is iteratively updated with the currently computed gradient $\nabla_{\mathbf{w}}Q(\mathbf{w}^{(k)})$ and some preset step sizes, $h_k$ ($k = 0, 1, 2, \cdots$), which are usually decaying.

The optimization in eq.(3) is clearly non-convex in the literal space for any nonlinear neural networks. However, due to the fact that both literal and canonical spaces are complete and the gradients in both

---

**Algorithm 1** Stochastic gradient descent (SGD) to learn neural networks in the literal space

---

    randomly initialize $\mathbf{w}^{(0)}$, and set $k = 0$
    **for** $epoch = 1$ **to** $L$ **do**
        **for** each $minibatch$ **in** training set $\mathcal{D}_T$ **do**
            $\mathbf{w}^{(k+1)} \leftarrow \mathbf{w}^{(k)} - h_k \nabla_{\mathbf{w}} Q(\mathbf{w}^{(k)})$
            $k \leftarrow k + 1$
        **end for**
    **end for**

---

spaces are linked through a point-wise linear transformation, in the following, we will theoretically prove that the gradient descent methods can solve this non-convex problem efficiently in the literal space in a similar way as solving normal convex optimization problems. As long as some minor conditions hold, the gradient descent methods surely converge to a global minimum even in the literal space.

**Theorem 2** *In the gradient descent methods in Algorithm 1, assume neural network is large enough ($M \geq N$), if the initial model, $\mathbf{w}^{(0)}$, and the step sizes, $h_k$ ($k = 0, 1, \cdots$), are chosen as such to ensure the disparity matrix $\mathbf{H}(\mathbf{w})$ maintains full rank at every $\mathbf{w}^{(k)}$ ($k = 0, 1, \cdots$), then it surely converges to a global minimum of zero loss. Moreover, the trajectory of $Q(\mathbf{w}^{(k)})$ ($k = 0, 1, \cdots$) behaves in the same as those in typical convex optimization problems.*

**Proof:** In non-convex optimization problems, as long as the objective function satisfies the Liptschitz condition and the step sizes are small enough at each step, the gradient descent in Algorithm 1 is guaranteed to converge to a stationary point, namely $\|\nabla_{\mathbf{w}} Q(\mathbf{w}^{(k)})\| \to 0$ as $k \to \infty$ (Nesterov, 2004). As long as $\mathbf{H}(\mathbf{w}^{(k)})$ maintains full rank, according to Lemma 1, $\mathbf{w}^{(k)}$ approaches a global minimum, i.e. $Q(\mathbf{w}^{(k)}) \to 0$ as $k \to \infty$.

Moreover, as shown in eq.(9), we have $\nabla_{\mathbf{w}} Q(\mathbf{w}^{(k)}) = \mathbf{H}(\mathbf{w}^{(k)}) \nabla_{\boldsymbol{\theta}} Q(\boldsymbol{\theta}^{(k)})$ for $k = 0, 1, \cdots$, where $\boldsymbol{\theta}^{(k)}$ denotes the corresponding representation of $\mathbf{w}^{(k)}$ in the canonical space. As $\nabla_{\mathbf{w}} Q(\mathbf{w}^{(k)}) \to 0$, $\mathbf{H}(\mathbf{w}^{(k)})$ normally does not approach 0. Let us show this by contradiction. Assume $\mathbf{H}(\mathbf{w}^{(k)}) \to 0$, which means that every row of $\mathbf{H}(\mathbf{w}^{(k)})$ approaches 0 at the same time. As we know, $m$-th row of $\mathbf{H}(\mathbf{w})$ corresponds to the Fourier series coefficients of the function $\frac{\partial f_{\mathbf{w}}(\mathbf{x})}{\partial w_m}$. If the Fourier coefficients are all approaching zero, it means that the function itself approaches 0 as well, i.e., $\frac{\partial f_{\mathbf{w}}(\mathbf{x})}{\partial w_m} = 0$. It means the current neural network function $f_{\mathbf{w}}(\mathbf{x})$ is irrelevant to the weight $w_m$. We may simply remove the link associated to $w_m$ without changing $f_{\mathbf{w}}(\mathbf{x})$. Because all rows of $\mathbf{H}(\mathbf{w})$ approaches 0, we may remove all links of all weights from the neural network without changing $f_{\mathbf{w}}(\mathbf{x})$. In all network structures used for artificial neural networks, after all weights are removed, there usually does not exist any link from input to output. As a result, as long as $\mathbf{H}(\mathbf{w}^{(k)}) \to 0$, the current $f_{\mathbf{w}}(\mathbf{x})$ approaches a constant function in the entire input range $\mathbb{U}_K$. This normally will not happen if a suitable initial model is chosen. Therefore, we have $\mathbf{H}(\mathbf{w}^{(k)}) \not\to 0$ for all $k = 0, 1, \cdots$. If all $\mathbf{H}(\mathbf{w}^{(k)})$ maintain full rank at the same time, as $\nabla_{\mathbf{w}} Q(\mathbf{w}^{(k)}) \to 0$, we will surely have $\nabla_{\boldsymbol{\theta}} Q(\boldsymbol{\theta}^{(k)}) \to 0$. The trajectory of $\boldsymbol{\theta}^{(k)}$ will converge towards a global minimum in the convex loss suface in the canonical space. Since the objective functions are equal across two spaces when we map $\mathbf{w}^{(k)}$ to $\boldsymbol{\theta}^{(k)}$, i.e., $Q(\mathbf{w}^{(k)}) = Q(\boldsymbol{\theta}^{(k)})$, then the trajectory of $Q(\mathbf{w}^{(k)})$ will go down steadily in the literal space until it converges to the global minimum. ∎

### 3.3 WHEN DO THE DISPARITY MATRIX $\mathbf{H}(\mathbf{w})$ DEGENERATE?

As shown above, the success of gradient descent learning in the literal space largely depends on the rank of the *disparity matrix* $\mathbf{H}(\mathbf{w})$. Here we will study under what conditions the *disparity matrix* $\mathbf{H}(\mathbf{w})$ may degenerate into a singular matrix. As we have seen before, the disparity matrix $\mathbf{H}(\mathbf{w})$ is a quantity solely depending on the structure and model parameters of neural network, $\mathbf{w}$, and it has nothing to do with the training data and the loss function. Since its columns are derived from orthogonal Fourier bases, its column vectors remain linearly independent unless its row vectors degenerate. As an $M \times N$ matrix ($M \geq N$), $\mathbf{H}(\mathbf{w})$ may degenerate in two different ways: i) some

rows vanish into zeros; ii) some rows are coordinated to become linearly dependent. In the following, we will look at how each of these cases may actually occur in actual neural networks.

### 3.3.1 DEAD NEURONS

In a neural network, if all in-coming connection weights to a neuron are chosen in such a way as to make this neuron remain inactive to the entire input range $\mathbb{U}_K$, e.g., choosing small connection weights but a very negative bias for a ReLU node. In this case, for any $\mathbf{x} \in \mathbb{U}_K$, this neuron does not generate any output. It becomes a *dead* neuron in the network. Obviously, for any connection weight $w$ to a dead neuron, $\frac{\partial f_\mathbf{w}(\mathbf{x})}{\partial w} = 0$. As a result, the corresponding row of $w$ in $\mathbf{H}(\mathbf{w})$ is all zeros. Therefore, dead neurons lead to zero rows in $\mathbf{H}(\mathbf{w})$ and they may reduce the rank of $\mathbf{H}(\mathbf{w})$. The bad thing is that the gradients of all weights to a dead neuron are always zero. In other words, the gradient descent methods can not save any dead neurons once they become dead. In a very large neural network, if a large number of neurons are dead, the model may not have enough capacity to universally approximate any function in $L^1(\mathbb{U}_K)$, this invalidates the assumption of completeness from the literal space to the canonical space. Obviously, the learning of this neural network may not be able to converge to a global minimum once a large number of neurons are dead.

### 3.3.2 DUPLICATED OR COORDINATED NEURONS

If many neurons in a neural network are coordinated in such a way that their corresponding rows in the disparity matrix $\mathbf{H}(\mathbf{w})$ become linearly dependent, this may reduce the rank of $\mathbf{H}(\mathbf{w})$ as well. However, in a large nonlinear neural network, the chance of such linear dependency is very small except the case of duplicated neurons. Two neurons are called to be duplicated only if they are connected to the same input nodes and the same output nodes and these input and output connection weights happen to be identical for these two neurons. Obviously, two duplicated neurons responds in the same way for any input $\mathbf{x}$ so that one of them becomes redundant. The rows of $\mathbf{H}(\mathbf{w})$ corresponding to duplicated neurons are the same so that its rank is reduced. Similar to dead neurons, the duplicated neurons will have the same gradients for their weights. As a result, once two neurons become duplicated, they will remain as duplicated hereafter in gradient descent algorithms. Like dead neurons, if there are a large number of duplicated neurons in a neural network, this may affect the model capacity for universal approximation. As a result, the learning may not be able to converge to a global minimum.

### 3.3.3 INITIAL CONDITIONS ARE THE KEY

In an over-parameterized neural network (assume $M \gg N$), the disparity matrix $\mathbf{H}(\mathbf{w})$ becomes singular only after at least $M - N$ neurons become dead or duplicated or coordinated, each of which essentially indicates all neural network parameters happen to satisfy an equality constraint. If the initial model $\mathbf{w}^{(0)}$ is randomly selected in such a way that all hyperplanes corresponding to all neurons intersect the input space $\mathbb{U}_K$ from the center as much as possible. The initial disparity matrix is nonsingular in probability one. When we use the gradient descent algorithms in Algorithm 1 to update the model, the chance to derive any new dead or duplicated neurons is extremely slim because it is unlikely for all parameters to simultaneously satisfy a large number of equality constraints. This is intuitively similar to the case where a large number of random points are generated in a high-dimensional space, the chance for all points to happen to lie in a hyper-plane is sufficiently small. Therefore, when an over-parameterized neural network is randomly initialized, the gradient descent algorithm in Algorithm 1 will converge to a global minimum of zero loss in probability one.

## 4 FINAL REMARKS

In this paper, we have presented some theoretical analysis to explain why gradient descents can effectively solve non-convex optimization problems in learning large-scale neural networks. We make use of a novel mathematical tool called *canonical space*, derived from Fourier analysis. As we have shown, the fundamental reason to unravel this mystery is the completeness of model space for the function class $L^1(\mathbb{U}_K)$ when neural networks are over-parameterized. The same technique can be easily extended to other function classes, such as linear functions (Baldi & Hornik, 1989; Kawaguchi, 2016; Haeffele & Vidal, 2015), analytic functions and band-limited functions (Jiang,

2019). As another future work, we may further quantitively characterize the disparity matrix $\mathbf{H}(\mathbf{w})$ to derive the convergence rate in learning large neural networks.

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
