# OpenReview forum: "Why Learning of Large-Scale Neural Networks Behaves Like Convex Optimization"
_ICLR.cc/2020/Conference — Reject_

### Official Review · AnonReviewer1 · 2019-10-25
**Official Blind Review #1**

**Rating:** 1

**Review:**

This paper claims to analyze the global convergence of large-scale neural networks (NNs). Their analysis relies on the idea of “canonical space”, which upon further reading, is nothing more than multi-dimensional Fourier analysis.

While the analysis of NNs is an extremely important problem, it is not clear what results are actually presented here nor how their results relate to other theoretical analyses of NNs. The introduction of Fourier analysis as a new mathematical tool, and devotion of a whole page to this “canonical space model” is almost laughable. Furthermore, this statement in the introduction is not only incorrect but it is absurd. To make such a claim would require a significant amount of background and review of literature to even place these statements in context.
        “No matter what structures are used in a large scale neural network, either feed-forward or recurrent, either convolutional or fully-connected, either ReLU or sigmoid, the simple first-order methods such as stochastic gradient descent and its variants can consistently converge to a global minimum of zero loss no matter what type of labelled training samples are used.”

Specific comments:
“to derive theoretical proofs under a very general setting without any unrealistic assumption on the model structure and data distribution.” - how is this possible?
“A new mathematical tool called canonical space” = Fourier analysis (??) this is not a new mathematical tool
Section 2 = a standard formulation of neural networks, followed by undergraduate level Fourier analysis.
In 2.1. What is a “well structured” network? Unclear from the notation and definitions.


**Experience Assessment:**

I have read many papers in this area.

**Review Assessment: Checking Correctness Of Derivations And Theory:**

I assessed the sensibility of the derivations and theory.

**Review Assessment: Checking Correctness Of Experiments:**

I assessed the sensibility of the experiments.

**Review Assessment: Thoroughness In Paper Reading:**

I read the paper at least twice and used my best judgement in assessing the paper.

---

### Official Review · AnonReviewer2 · 2019-10-27
**Official Blind Review #2**

**Rating:** 1

**Review:**

This paper looks at the neural net training problem in a "canonical space" which is parameterized by the Fourier coefficients of the function. This canonical space is a bijection of the function space L^1([0, 1]^K), and if we allow an epsilon approximation of the function we can truncate the Fourier coefficients so that the canonical space is finite-dimensional. The paper shows that in the canonical space, the training problem is always convex. Going back to the literal space (original parameter space for a neural network), it is shown that as long as a "disparity matrix" remains full rank, gradient descent will converge to a global minimum.

I don't think this paper has anything new or non-trivial. I also don't think it's helpful to look at the canonical space proposed in the paper. In particular, it just transfers the difficulty of the problem into a disparity matrix, which we actually don't have control over. The paper claims that the matrix can be made full rank. This is not correct. Maybe one can prove it's full rank at random initialization, but I don't see how to prove this throughout training. The authors would need to provide a rigorous proof in order to claim this.

In fact, in non-convex optimization it's easy to arrive at a scenario where you "only" need some matrix to remain full rank in order to prove convergence to global minimum. One such example is the recent series of work on neural tangent kernel (NTK). There, as long as the NTK matrix stays full rank (actually, one needs eigenvalues bounded away from 0), one can show convergence to global minimum. However, to actually show this, one needs to apply stringent assumptions on the neural network architecture and to devote dozens of pages to the proof.

**Experience Assessment:**

I have published in this field for several years.

**Review Assessment: Checking Correctness Of Derivations And Theory:**

I assessed the sensibility of the derivations and theory.

**Review Assessment: Checking Correctness Of Experiments:**

N/A

**Review Assessment: Thoroughness In Paper Reading:**

I read the paper at least twice and used my best judgement in assessing the paper.

---

### Official Review · AnonReviewer3 · 2019-10-28
**Official Blind Review #3**

**Rating:** 1

**Review:**

Summary:
This paper presents an argument that massively overparameterized neural networks trained with gradient descent on a supervised learning problem with convex loss function will converge. It argues that by mapping each model to a truncated Fourier series we can recover a canonical representation for the functions in the model space. Viewing this as a linear map at each point in parameter space and assuming that this matrix is always full rank, they claim convergence of gradient descent.


Decision:
This paper should be rejected because it lacks originality, the assumptions are often too strong, and the rigor of some claims is questionable.

Main argument:
Originality:
This is the main issue with the paper. Several papers, for example the ones from Allen-Zhu et al. and Du et al. cited in the paper, have shown that gradient descent on supervised learning provably converges for massively overparameterized neural networks. Not only that, but those works give rigorous proofs of their claims and even handle issues like step size, convergence rates, and precise conditions on the size and architecture of the neural network. This paper does not attempt to handle any of these details, claiming to simplify the proof. To me the proof is only simplified since the details are all omitted.

Assumptions:
The main way that details are avoided is by adding a strong assumption that the “disparity matrix” remains full rank over the course of training. This assumption is not supported in an empirical or formal way, but rather by appeal to some vague argument in section 3 that since the matrix is likely to be full rank at initialization, it is likely to remain so for large enough networks.

Rigor:
As explained above, many details are omitted in favor of strong assumptions, but there are also some technical details that I think may be wrong. On page 2, it is claimed that the mapping from Lambda_M to L^1 is surjective, this is not true. Every function in L^1 can be approximated by some function in Lambda^M, but for finite M this map cannot be surjective. Another similar issue arises in the proof of Theorem 1 when it is claimed that there is a unique theta_epsilon corresponding to each f. This is again false since by truncating the Fourier series, infinitely many functions will get mapped to the same truncation (when they only differ in the higher coefficients). These issues make me question the validity as well as the originality of the arguments presented.


Additional feedback:
- I saw a few spelling and grammatical errors. For example: in the abstract the tenses alternate between present and past almost every sentence (“we have proved” should be “we prove”),  page 2 “parametarize”, page 3 “serie”, page 5 “distint”, page 7 “Liptschitz”, section 3.3 title should be “when does” not “when do”.

**Experience Assessment:**

I have read many papers in this area.

**Review Assessment: Checking Correctness Of Derivations And Theory:**

I assessed the sensibility of the derivations and theory.

**Review Assessment: Checking Correctness Of Experiments:**

N/A

**Review Assessment: Thoroughness In Paper Reading:**

I made a quick assessment of this paper.

---

### Official Review · AnonReviewer4 · 2019-10-31
**Official Blind Review #4**

**Rating:** 1

**Review:**

The paper studies the problem of optimization for neural networks. It compares the optimization problem in parameter space with the corresponding problem in function space. In particular, it parameterizes function space using Fourier coefficients, so that the new problem is convex. When the "disparity matrix" (Jacobian) of the mapping from parameter space into function space has full rank, then critical points in parameter spaces are critical points in function space and hence global minima. The paper concludes by stating that "an over-parameterized neural network is randomly initialized [...] will converge to a global minimum of zero loss in probability one".

I believe that the paper in its current form should be rejected. The main reason is that the second part of the paper is not rigorous, and the results that are shown do not imply that the optimization process will converge to a global optimum (contrary to what is explicitly stated). Indeed, while it is reasonable to assume that the "disparity matrix" should have full rank at initialization, this may not remain true throughout the training process.
There are also other technical problems, discussed below.

Some comments:

* The notation Q(.) is used inconsistently (Q(f), Q(w) and Q(\theta)). The same is true for f_*, (f_w and f_\theta).
* I believe the disparity map is simply the Jacobian of the mapping w->theta.
Also, there are missing gradient symbols in eq. 8.
* The "canonical model space" is not clearly defined. I believe it is simply L1(U) parameterized in a certain way, but this should be stated.
* By considering the mapping into the truncated "canonical model space" (i.e., using a finite number of Fourier coefficients) then f_theta is not exactly equal to f_w. In particular, if f_theta has zero loss, then the same is not necessarily true for f_w. Thus, we cannot conclude that f_w is a global minimum.
* The last statement in Theorem 2 is not clear ("the trajectory [...] behaves in the same as those in typical convex optimization problems"). From the proof, I believe it should mean that the disparity map never vanishes, but I don't understand why this is relevant (we would like for it to have full rank).
* The main result of the paper rests on the claim "When we use the gradient descent algorithms in Algorithm 1 to update the model, the chance to derive any new dead or duplicated neurons is extremely slim because it is unlikely for all parameters to simultaneously satisfy a large number of equality constraints" but this is not rigorous and wrong.
* Some typos: "pointwise distint", "suface"

**Experience Assessment:**

I have published one or two papers in this area.

**Review Assessment: Checking Correctness Of Derivations And Theory:**

I assessed the sensibility of the derivations and theory.

**Review Assessment: Checking Correctness Of Experiments:**

N/A

**Review Assessment: Thoroughness In Paper Reading:**

I read the paper at least twice and used my best judgement in assessing the paper.

---

### Decision · Program_Chairs · 2019-12-19

**Decision:**

Reject

**Comment:**

This paper studies the problem of optimization for neural networks, by comparing the optimization problem in parameter space with the corresponding problem in function space. It argues that overparametrised models leads to a convex problem formulation leading to global optimality.

All reviewers agreed that this paper lacks mathematical rigor and novelty relative to the current works on overparametrised neural networks. Its arguments need to be substantially reworked before it can be considered for publication, and as a consequence the AC recommends rejection.